# Linear Irreversible Thermodynamics: A Glance at Thermoelectricity and the Biological Scaling Laws

**DOI:** 10.3390/e25121575

**Published:** 2023-11-23

**Authors:** Juan Carlos Chimal-Eguia, Ricardo Teodoro Páez-Hernández, Juan Carlos Pacheco-Paez, Delfino Ladino-Luna

**Affiliations:** 1Laboratorio de Ciencias Matemáticas y Computacionales, Centro de Investigación en Computacion, Instituto Politecnico Nacional, Ciudad de Mexico 07738, Mexico; 2Area de Fisica de Procesos Irreversibles, Departamento de Ciencias Basicas, Universidad Autonoma Metropolitana, U-Azcapotzalco, Av. San Pablo 180, Col. Reynosa, Ciudad de Mexico 02200, Mexico; dll@azc.uam.mx; 3Departamento de Ciencias Basicas, Universidad Autonoma Metropolitana-Azcapotzalco, Ciudad de Mexico 02200, Mexico; jcpacheco@azc.uam.mx

**Keywords:** linear irreversible thermodynamics, thermoelectricity, scaling laws

## Abstract

This paper presents so-called thermoelectric generators (TEGs), which are considered thermal engines that transform heat into electricity using the Seebeck effect for this purpose. By using linear irreversible thermodynamics (LIT), it is possible to study the thermodynamic properties of TEGs for three different operating regimes: maximum power output (MPO), maximum ecological function (MEF) and maximum power efficiency (MPE). Then, by considering thermoelectricty, using the correspondence between the heat capacity of a solid and the metabolic rate, and taking the generation of energy by means of the metabolism of an organism as a process out of equilibrium, it is plausible to use linear irreversible thermodynamics (LIT) to obtain some interesting results in order to understand how metabolism is generated by a particle’s released energy, which explains the empirically studied allometric laws.

## 1. Introduction

Although many physical phenomena were observed during the 19th century, many of them did not have a sufficiently clear explanation based on first principles of physics, nor were they completely described in a mathematical language. It is in this context that in 1821, Seebeck [1] published his novel idea: the generation of an electric current using the thermodynamic properties of the involved materials. This marked a new way to understand thermoelectric phenomena and their implications. This idea has since paved the way for other researchers to develop other similar ideas in the nascent field. Without any doubt, among the most important is that studied by Peltier in 1834 [2], according to which a temperature gradient develops from the flowing electrical current, as well as the observations made by Thomson [3], according to which he proposed that in order to heat or cool an electrical conductor, we only need a difference in temperature at the ends of the conductor.

It is very interesting to note that it was the same W Thomson (Lord Kelvin) who, in 1854, began the study of irreversible processes out of equilibrium, formally analyzing the phenomenon of thermoelectricity. Later, RJ Clausius [4,5] and J. C. Maxwell [5] gave greater solidity to the ideas of irreversibility, culminating in 1872, with Ludwig Boltzmann [5], who proposed the equation that bears his name and established the first theory for irreversible processes.

Now, it is well known that the fundamental relationship in classical thermodynamics postulates that a change in the entropy of a system is a function of chemical potential, pressure, and—obviously—temperature (called intensive quantities). In 1931, Onsager et al. [6] proposed the extension of this relationship to non-equilibrium thermodynamic systems, for which they considered some new locally defined intensive macroscopic variables. Under the right conditions, Onsager et al. were able to obtain these new variables by defining the gradients and flux densities of the defined basic macroscopic quantities. Such gradients, now known as "thermodynamic forces", produce flux densities or "fluxes" that are in duality with the forces. These quantities are defined by the Onsager reciprocity relations and form the foundations of what, years later, would be called linear irreversible thermodynamics (TIL).

On the other hand, after many years of dormancy for classical equilibrium thermodynamics (CET), in the second half of the 20th century, with the pioneering works of Curzon and Ahlborn, newborn finite-time thermodynamics (FTT) led to one of the most successful models to describe heat engines under more realistic conditions [7,8,9,10,11]. In this sense, CET had only shown what the upper limits were for some variables related to the process itself, such as the efficiency.

Thus, by considering the paper by Curzon and Ahlborn [12], many authors began to introduce different objective functions, such as the ecological function [13], the omega function [14], the efficient power function [15], etc. These researchers were mainly attempting to obtain efficiency and power values for real thermal engines [16,17,18].

Moreover, in 1965, Kedem et al. [19] published the first step of a non-equilibrium theory towards a description of linear converters of energy (which would be called linear irreversible thermodynamics (LIT)). Since then, many authors have considered this theory as a basis for the analysis of non-equilibrium systems, particularly for biological processes; several authors, have studied different optimal regimes, like Prigogine [20], with his minimum entropy production theorem. Odum and Pinkerton [21], who analyzed the maximum power output regime for various biological systems, and Stucki et al. [22], who introduced some optimal criteria to study the optimum oxidative phosphorylation regime, are among others in this area [23,24,25,26,27] who have studied many biological energy conversion processes by means of LIT, analyzing optimum performance regimes.

Within this context, if thermoelectric generators (TEGs) are considered thermal engines that directly convert heat into electricity via the Seebeck effect, it is possible to study the thermodynamic properties of TEGs using linear irreversible thermodynamics (LIT) for this purpose for three different operating regimes: maximum power output (MPO), maximum ecological function (MEF) [13], and maximum efficient power function (MEPF) [15].

It is important to note that in a very simple way, the use of LIT allows the maximum electric current ($I_{max}$) and the maximum power output ($P^{max}$) to be obtained for the three aforementioned regimes. Although the maximum electric current for the maximum power output regimes has been obtained and most work has been carried out using this regime, the maximum electric current for the other two work regimes has not been obtained, much less the expressions of the maximum output power. There exist more sophisticated models; for instance, the fluid-thermal-electric multiphysics numerical model, in which the performance of a thermoelectric generator is obtained by considering the complete geometry, temperature-dependent material properties, and topological connection among the thermoelectric modules [28], which can be used to compare the results presented here with those of practical devices.

The latter is important because when an optimal thermoelectric device is going to be designed in practice. One could take into account the geometric properties of the materials in order to gain more power or, depending on the goal being pursued, maximum power; then, we can use the maximum power regime, i.e., a balance between entropy production and output power, and the maximum ecological function [13]. Finally, if we need to consider the effects on the design of the device, as well as the multiplication of power by the cycle efficiency, then the maximum efficient power criterion [15] has to be used.

Finally, we intend to show a very interesting link between the themoelectricity phenomena and the very well-studied allometric laws [29,30,31]. Many researchers have described the fact that there is a relationship between the rate at which an individual expends energy and their body mass in adulthood expressed mathematically using a power law. Among the biological studies, there is an agreement with respect to the exponents of the power law but not with respect to the constants of proportionality of these allometric laws. Therefore, by using a correspondence between the heat capacity of a solid at a low temperature and the metabolic rate, taking the generation of metabolic energy as a non-equilibrium process, it is plausible to use linear irreversible thermodynamics (LIT) to obtain different proportionality constants for the mentioned allometric laws; then, we can assume that there exist different constants of proportionality because they are associated with different work regimes under which different organisms might work, thinking of them as thermodynamic machines, a phenomenon that has not been explained by biologists.

## 2. Thermodynamics of Thermoelectric Phenomena: A Brief Overview

### 2.1. Fluxes and Thermoelectric Coefficients

We can place the birth of the study of thermodynamics of thermoelectric phenomena at the beginning of the 19th century, with the pioneering works of Thomas J. Seebeck [1]. Since then, very interesting work has been carried out in the field [32,33,34,35,36,37,38,39,40,41,42]. In particular, we can summarize findings obtained by describing thermoelectric phenomena using linear irreversible thermodynamics [25,32,41] as follows.

Let us suppose that thermoelectric materials and classical thermal engines can both be visualized as composed of a perfect gas. For the thermal engine, we can assume that we have a molecular gas, and for the thermoelectric material, we can assume a gas made up of electrons. Now, as in classical thermodynamics, let us consider that one end of a thermoelectric sample is maintained at a temperature of $T_{h}$ and that the other is maintained at a temperature of $T_{c}$, where $T_{h}>T_{c}$. Since heat flows from the hot to the cold end, the system cannot be considered to be under equilibrium conditions. Now, thinking of the particles in the gas as charged particles, we have an electrochemical difference (usually called voltage difference), which is obtained by considering the difference in the temperatures; seen another way, there is a coupling between the electrochemical potential and a difference in the temperatures (a gradient of temperatures). Then, when we consider problems near equilibrium, i.e., small perturbations, one can assume linear coupling between the forces and fluxes.

By following the Onsager formalism [6], we can establish a relationship between such forces and fluxes near the steady thermodynamically non-equilibrium regime naming them phenomenological relations, as expressed by:(1)$$J_{\delta }=\sum_{\delta }L_{\delta \psi }X_{\psi }$$
where $L_{\delta \psi }$ represents the phenomenological coefficients, usually depending on the intensive variables that describe the coupling between two irreversible processes (δ and ψ), and $X_{\psi }$ represents the respective thermodynamic forces. It is worthwhile to mention that in 1931, Onsager [6] demonstrated that for a system of flows and forces based on an appropriate dissipation function, the matrix of coefficients is symmetrical so that the phenomenological coefficients have the following symmetry relation: $L_{\delta \psi }$ = $L_{\psi \delta }$, which affords a considerable reduction in the number of measured coefficients.

Then, by taking the above into account, it is possible for our system to establish two thermodynamic flows, i.e., $J_{particles}$ and $J_{heat}$ (the analysis of the expressions of these potentials is outside the scope of the present article, but we refer the reader to the discussions in [32,43]), for which we can write the following phenomenological equations (see Equation (Equation 4) proposed by Goupil et al. [32]):(2)$$\left[\begin{array}{c} J_{\mathrm{particles}}\\ J_{\mathrm{heat}} \end{array}\right]=\left[\begin{array}{cc} L_{11} & L_{12}\\ L_{21} & L_{22} \end{array}\right]*\left[\begin{array}{c} -\frac{1}{T}\nabla \mu_{e}\\ \nabla (\frac{1}{T}) \end{array}\right]$$
where $J_{heat}$ is the heat flux, $J_{particles}$ is the particle flux, $\mu_{e}$ is the electrochemical potential, and *T* is the temperature of a gas made up of electrons. For the heat flux and the particle flux, there exist relations with their respective thermodynamic potential gradients; for the particle flux, the associated gradient is $X_{N}=\frac{-1}{T}\nabla \mu_{e}$, and for the heat flux, the corresponding gradient is $X_{heat}=\nabla \left(\frac{1}{T}\right)$.

It is worth mentioning that since the electric field is derived from the electrochemical potential, we can derive the following relation: (3)$$E=\frac{-\nabla \mu_{e}}{e}=-\nabla V$$
where *e* is the particle’s charge (for our systems, *e* is the charge of the electron) and *V* is the electrical potential.

One very interesting result is obtained when, under certain thermodynamic conditions, thermoelectric coefficients $L_{11},\;L_{12},\;L_{21}$, and $L_{22}$ are derived from the fluxes. Then, by using Equation (Equation 2) under isothermal conditions, it is possible to obtain the electrical conductivity ($\sigma_{T}$), as well as the thermal conductivity ($k_{E}$) and the Seebeck coefficient (α) as [32];
(4)$$\sigma_{T}=\frac{e^{2}}{T}L_{11}$$
(5)$$k_{E}=\frac{L_{22}}{T^{2}}$$
(6)$$\alpha =\frac{1}{eT}\frac{L_{12}}{L_{11}}$$

It is worth mentioning that there exists a closed relation between the Peltier effect and the Seebeck effect by means of the α coefficient, since Π=αT, where Π is the Peltier coefficient.

### 2.2. Some Insights on Linear Irreversible Thermodynamics

In an attempt to study the important problem of energy transfer in biological systems, Caplan et al. [24] investigated some linear energy converters working in steady states; they introduced the thermodynamic definitions of power output and efficiency, as well as the known notion of entropy production rate, using linear irreversible thermodynamics (LIT) for this purpose. With Caplan’s definitions in mind, Stucki et al. [22] analyzed some optimum regimes different from that of minimum entropy production previously studied by Prigogine et al. [20]. Since then, it has become clear that in many systems (biological, physical, chemical, etc.), the study of some optimum working regimes is important in order to understand the diverse ways in which energy can be transferred.

From this perspective and taking into consideration the results obtained by Caplan and others [24,25,26] for TIL, we can summarize some findings on the thermodynamic properties of different work regimens.

#### Maximum Power Output

When we study some systems, considering linear energy converters and linear irreversible thermodynamic techniques, it is possible to obtain the power output and the efficiency of the system in non-reversible states that are in contact with one thermal source as [24,25];
(7)$$P=-TJ_{1}X_{1}$$
and
(8)$$\eta =-\frac{J_{1}X_{1}}{J_{2}X_{2}}$$
where T is the temperature of the thermal bath, and $J_{1}$, $J_{2}$ and $X_{1}$, $X_{2}$ are the fluxes and the forces (called phenomenological relations), respectively, with the following linear relation: (9)$$J_{1}=L_{11}X_{1}+L_{12}X_{2}$$
(10)$$J_{2}=L_{21}X_{1}+L_{22}X_{2}$$
which are exactly the same relations shown in Equation (Equation 2) for the thermoelectric phenomena.

It is well established that by using Equations (7)–(11), it is possible to obtain the power output, efficiency, and entropy generation as [25,27,41],
(11)$$P=TL_{22}X_{2}^{2}q^{2}x\left(1-x\right)$$
(12)$$\eta =\frac{q^{2}x\left(1-x\right)}{\left(1-q^{2}x\right)}$$
and
(13)$$\sigma =L_{22}X_{2}^{2}\left[\left(1-q^{2}\right)+q^{2}{\left(1-x\right)}^{2}\right]$$
respectively, where $q=\frac{L_{12}^{2}}{L_{11}L_{22}}$ is the degree of coupling (it is worth mentioning that the value of *q* is related to what, in thermoelectricity, we call the figure of merit (*Z*)), and $x=-\frac{L_{11}}{L_{12}}\frac{X_{1}}{X_{2}}$ is the driven force.

Once we can obtain the power output, efficiency, and entropy production (σ) one important question that arises is whether the systems work in different performance regimes in order to optimize either power output, entropy production, or a mixture of these quantities. The above was carried out in an effort understand why certain systems generate efficiencies that are not obtained when using classical thermodynamics.

Among the most cited results in the literature, we found four that seem to be in agreement with the observations made in real systems, which are listed as follows:Maximum power output (MPO):For this regime, when we maximize the power function, i.e., $\frac{\partial P}{\partial x}=0$, the maximum is obtained when x=0.5. Thus, if we substitute this value into Equations (11)–(13), we obtain the power output, efficiency, and entropy production, respectively, which have the the following expressions:
(14)$$P_{MPO}=TL_{22}X_{2}^{2}\left(\frac{1}{4}q^{2}\right),$$
(15)$$\eta_{MPO}=\frac{1}{2}\frac{q^{2}}{\left(2-q^{2}\right)},$$
and
(16)$$\sigma_{MPO}=L_{22}X_{2}^{2}\left(1-\frac{3}{4}q^{2}\right)$$Maximum entropy production (MEP):For this regime, when we maximize the entropy production, i.e., $\frac{\partial \sigma }{\partial x}=0$, the maximum is obtained when x=1. Thus, if we substitute this value into Equations (11)–(13), we obtain the power output, efficiency, and entropy production, respectively, which have the following expressions:
(17)$$P_{MEP}=0$$
(18)$$\eta_{MEP}=0$$
and
(19)$$\sigma_{MEP}=L_{22}X_{2}^{2}\left(1-q^{2}\right)$$Maximum ecological function (MEF):For this regime, when we maximize the ecological function (E=P−Tσ), i.e., $\frac{\partial E}{\partial x}=0$, the maximum is obtained when x=0.75. Thus, if we substitute this value into Equations (11)–(13), we obtain the power output, efficiency, and entropy production, respectively, which have the following expressions:
(20)$$P_{MEF}=TL_{22}X_{2}^{2}\left(\frac{3}{16}q^{2}\right),$$
(21)$$\eta_{MEF}=\frac{3}{4}\frac{q^{2}}{\left(4-3q^{2}\right)},$$
and
(22)$$\sigma_{MEF}=L_{22}X_{2}^{2}\left(1-\frac{15}{16}q^{2}\right)$$Maximum efficient power (MEP):For this regime, when we maximize the efficient power function ($P_{E}=\eta P$), i.e., $\frac{\partial E}{\partial x}=0$, the maximum is obtained when x=0.6 (this result is only yielded for the case in which q=1 for a different *q* value, and the value of *x* is given by $x=\frac{4+q^{2}-\sqrt{16-16q^{2}+q^{4}}}{6q^{2}}$). Thus, if we substitute this value into Equations (11)–(13), we obtain the power output, efficiency, and entropy production, respectively, which have the following expressions:
(23)$$P_{MEP}=\frac{2}{9}TL_{22}X_{2}^{2}$$
(24)$$\eta_{MEP}=\frac{2}{3}$$
and
(25)$$\sigma_{MEP}=\frac{1}{9}L_{22}X_{2}^{2}$$

### 2.3. Linear Irreversible Thermodynamics and the Thermoelectric Generator

We have analyzed the concepts of TIL and electricity. Now, we are able to describe some thermodynamic properties of thermoelectric phenomena using linear irreversible thermodynamics, focusing on the way in which some thermoelectric devices (e.g., a thermoelectric generator) transforms energy by considering three different working regimes, namely (a) maximum power (MPO), (b) maximum ecological function (MEF), and (c) maximum power efficiency (MPE).

#### 2.3.1. Maximum Power Output Regime

According to Equation (Equation 7), the power output is given by:(26)$$P=-TJ_{particles}X_{particles}$$

Then, using Equation (Equation 2), it is possible to transform Equation (Equation 26) as follows:(27)$$P=-T\left[L_{11}\left(-\frac{1}{T}\nabla \mu_{e}\right)+L_{12}\nabla \left(\frac{1}{T}\right)\right]\left[-\frac{1}{T}\nabla \mu_{e}\right]$$

Now, substituting Equations (3), (4), and (6) into Equation (Equation 27) yields:(28)$$P=[-\sigma_{T}E+\sigma_{T}\alpha \nabla T]E$$

We must remember that the electric field (*E*) is related to the electrical potential (*V*) and that in real electrical devices (for example, in a thermogenerator [32]), we can relate this electrical potential to an electrical current (*I*) by means of Ohm’s law as followings: (29)$$I=\frac{V_{0}}{R_{in}}$$
where $R_{in}$ is the internal resistance of the electrical device.

Thus, by substituting Equation (Equation 29) into Equation (Equation 28), it is possible to obtain:(30)$$P=-\sigma_{T}R_{in}\left[I^{2}R_{in}-I\alpha \nabla T\right]$$

It is important to note that the internal Ohmic resistance ($R_{in}$) is related to the electrical conductance ($\sigma_{T}$) by the following relation: $R_{in}=\frac{L}{\sigma_{T}A_{c}}$ [32], where *L* is the length of the material and $A_{c}$ is the cross-sectional area of a certain thermoelectric material. Taking into account the above, Equation (Equation 30) transforms as:(31)$$P=\frac{L}{A_{c}}\left[-I^{2}R_{in}+I\alpha \nabla T\right]$$

We note that this last equation is a function of the current (*I*), so if we can obtain the maximal current for the device, then it is possible to obtain the maximal electric power output.

Then, in order to obtain the maximal electrical current, we need to remember that according to Equation (Equation 13), the driven force (*x*) is given by:(32)$$x=-\frac{L_{11}}{L_{12}}\frac{X_{1}}{X_{2}}$$

So, substituting Equations (2), (4), (5), (7), and (29) into Equation (Equation 32) yields:(33)$$x=-\frac{IR_{in}}{\alpha \nabla T}$$

We have shown that under the MPO regime, the power output reaches its maximum when $x=\frac{1}{2}$; therefore, we can substitute the above value into Equation (Equation 33), obtaining:(34)$$\frac{1}{2}=-\frac{IR_{in}}{\alpha \nabla T}$$

When we substitute x=0.5 into Equation (Equation 33), we obtaining the maximum power output of the device. This happens when we have a maximum electrical current ($I_{max}$); thus,
(35)$$I_{max}=\frac{\alpha \nabla T}{2R_{in}}$$

If we take into consideration that the open voltage of a thermoelectric generator is given by $V_{0}=\alpha \nabla T$, substituting $V_{0}$ into Equation (Equation 35) yields:(36)$$I_{max}=\frac{V_{0}}{2R_{in}},$$
which is the classical maximum electrical current obtained by Goupil et al. [32] and others for a thermoelectric generator but using a different approach.

Finally, using Equation (Equation 31), it is easy to obtain the maximum power output in terms of the maximal electrical current ($I_{max}$), only substituting Equation (Equation 36) into Equation (Equation 31), obtaining the following expression:(37)$$P_{MPO}^{max}=\frac{\alpha^{2}\nabla T^{2}}{4}\sigma_{T}$$

Again, this is a classical result for a thermoelectric generator (see Equation (Equation 49) in [32]). It is important to note that the same result could have been obtained directly using Equation (Equation 14) and substituting Equations (2) and (6) and considering that electrical conductivity must be greater than the thermal conductivity and that the coupling factor should be q=1.

#### 2.3.2. Maximum Ecological Function Regime

Following the same ideas as in Section 2.3.1, we can obtain the maximal electric current and the maximal power output but for the ecological function regime. In the context of the finite-time thermodynamics the regime, the ecological working regime [13] consists of the maximization of a function (*E*) that represents a relationship between high-power output and low-entropy production per cycle. This function is given by E=P−Tσ, where *P* is the power output and σ the total entropy production (system plus surroundings) and *T* is the temperature of the cold reservoir.

Then, in order to obtain the maximal electrical current, we need to remember that the driven force (*x*) is given by $x=-\frac{IR_{in}}{\alpha \nabla T}$. We also showed that under the MEF regime, the power output reaches its maximum when $x=\frac{3}{4}$; therefore, we can substitute the above value into Equation (Equation 33), obtaining:(38)$$I_{max}=\frac{3V_{0}}{4R_{in}}$$
which is the maximum electrical current for the thermoelectric generator in the ecological function regime. Moreover, according to Equation (Equation 31), it is easy to obtain the maximum power output for the ecological function regime in terms of the maximal electrical current ($I_{max}$), only substituting Equation (Equation 38) into Equation (Equation 31), obtaining the following expression:(39)$$P_{MEF}^{max}=\frac{3\alpha^{2}\nabla T^{2}}{16}\sigma_{T}$$

As we mentioned before, the same result could have been obtained directly using Equation (Equation 20) and substituting Equations (2) and (6), as well as considering that the electrical conductivity must be grater than the thermal conductivity and that the coupling factor should be q=1.

#### 2.3.3. Maximum Efficient Power Regime

Following the same ideas as in Section 2.3.1 and Section 2.3.2, we can obtain the maximal electric current and the maximal power output but for the efficient power regime. The maximum efficient power regime given by Yilmaz et al. [15] considers the effects on the design of heat engines, as well as the power and the cycle’s efficiency. This criterion has been successfully applied to the Carnot, Brayton, and diesel engines, among other systems. Therefore, the approach called maximum efficient power in the context of thermal engines is defined as $P_{e}=\eta P$, where *P* is the power output and η is the efficiency of the cycle. Maximization of this function provides a compromise between power and efficiency, where the designed parameters under maximum efficient power conditions lead to more efficient engines than those under the maximum power conditions [15].

Then, in order to obtain the maximal electrical current, we showed that under the MEP regime, the maximum power output reaches its maximum when $x=\frac{3}{4}$. Then, we can substitute the above value into Equation (Equation 33), obtaining:(40)$$I_{max}=\frac{2V_{0}}{3R_{in}}$$

Which is the maximum electrical current for a thermoelectric generator under the efficient power regime. Moreover, according to Equation (Equation 40), it is easy to obtain the maximum power output for the efficient power regime in terms of the maximal electrical current ($I_{max}$), only substituting Equation (Equation 40) into Equation (Equation 31), obtaining the following expression:(41)$$P_{MEF}^{max}=\frac{2\alpha^{2}\nabla T^{2}}{9}\sigma_{T}$$

As we mentioned before, the same result could have been obtained directly using Equation (Equation 23) and substituting Equations (2) and (6), as well as considering that the electrical conductivity must be grater than the thermal conductivity. Figure 1 depicts the maximum power outputs for the three different regimes under study.

Finally, Figure 2 depicts the maximum electrical current for the three different regimes considered in this paper.

## 3. Allometric Laws

Now, we tackle a problem that is, in principle, totally different from what we have been talking about. The aforementioned problem involves finding the origin of empirical laws in animal physiology that relate the energy consumption rate or metabolic rate to the size of organisms, known as allometric scaling laws [29,30,31].

### 3.1. Introduction to the Allometric Laws

Since the mid-20th century, several authors have delineated these empirical laws based on empirical studies. Of particular relevance are the studies conducted by Kleiber et al., Brody et al., and Hemmingsen et al. [29,30,31]. All these researchers described the fact that there is a relationship between the rate at which an individual expends energy under thermoneutral conditions and their body mass in adulthood. This relation is usually expressed in terms of two parameters: a scaling exponent and the normalizing coefficient. For large mammals, the scaling exponent has been placed around 0.75, and for small mammals and birds, the exponent has been placed around 0.66 [44,45]. A normalizing coefficient has been demonstrated that has a large range of values depending on biological features, from plants to animals, including unicellular organisms.
(42)$$P=AW^{\beta }$$
where *P* is the basal metabolic rate, *A* is the normalizing coefficient, *W* is the body mass, and β is the scaling exponent.

The foregoing is the product of phenomenological observations; however, at the cellular level, the organization and synthesis of molecules that allow for the continuous supply of energy to the interior of the cells are required. It is extremely important that there is a mechanism that allows these molecules to be transported into cells without compromising cell integrity, making their transduction selective and orchestrated from the cell membrane itself. The biolipid layer that forms the cell membrane, as well as the inner membrane in the mitochondria, uses similar mechanisms (even in plants, the organization and functioning are similar); that is, without the phosphorylation of ADP through the structure of the biomembranes, it would not be possible to carry out the release of energy through the oxidation of ADP, which is coupled with proton translocation through the membrane.

In other words, the number of positive charges (protons) that cross the cell membrane is called the metabolic flux. These charges are released due to the coupling between electron transport and ADP phosphorylation. We denote this flux as *J*. At the same time that protons are transported into the cell, an electrochemical gradient is produced, called the proton motive force (Δp).

Thus, if we denote *C* as the conductance of the membrane and *J* as the current of protons (produced by the electromotive force due to the transport of electrons), then by simply using Ohm’s law, we obtain J=CΔp.

Regarding this last expression, the process of energy transfer determines a metabolic flux that, according to Demetrius et al. [46], defines the metabolic energy per particle generated per cycle, as expressed by E=Jτ, where τ denotes the cycle time, i.e., the mean turnover time for the redox reactions. This characterization of the metabolic energy of the molecule can be used in order to determine the total metabolic energy generated by downhill electron transfer during one cycle of the redox reaction in the following manner. By taking into account Equation (Equation 42), a multicellular organism of size *W* can be considered an aggregate of closely packed identical cells. This means that an organ or a tissue can be considered as the sum of all cellular metabolic rates, obtaining a total empirical relation representing the total metabolic rate as [47]:(43)$$P=\gamma C\Delta pW^{\beta }$$
where γ denotes the efficiency with which energy is transported between the different cells and tissues within the organism.

Finally, it is worth mentioning that there exist two analytic expressions for the scaling exponent (β)as a function of metabolic efficiency (μ), namely β=(2μ−1)/μ and β=(4μ−1)/4μ, which, for each case, specify the class of organisms to which these different expressions of β refer, i.e, perennial plants (β=1) or mammals and large birds (β=2/3).

### 3.2. Metabolic Energy as a Non-Equilibrium Process

As we previously mentioned, metabolic energy is the result of the excitation of molecules, which drives a downhill flow of electrons due to the coherent excitation of the molecular groups. Based on this simple assumption, Demetrius et al. [46] proposed so-called “quantum metabolism", which assumes that Planck’s quantization of radiation oscillators can also be applied to the vibration of large molecular groups embedded in the membrane, where energy transduction occurs on the microscopic level (i.e., the coupling of two molecular motors: the redox reaction and ADP phosphorylation, in which the structures where the cycle happens consist of lipid–protein complexes, which are embedded in the phospholipid bilayer). Then, this is all held together by many cooperative non-covalent interactions, where the embedded proteins have dipolar properties and can be expected to exhibit oscillations. Moreover, we can assume that the metabolic energy of the system is determined by 3N harmonic molecular oscillators.

Thus, it is plausible to consider Debye’s quantum theory of solids, which postulates that the material oscillators are not actually independent of the oscillations of the others but are coupled with them because of the forces between the molecules, which describes the mechanism of energy transduction in terms of the collective modes of vibration of molecular groups using a correspondence between the heat capacity of a solid at a low temperature and the metabolic rate and taking the generation of metabolic energy as a non-equilibrium process; therefore, it is plausible to use linear irreversible thermodynamics (LIT) to obtain some interesting results in order to understand how metabolism is generated by the energy released by the particles as they are transferred from the donor to acceptor states within the energy-transducing biomembrane.

### 3.3. The Heat Capacity and Its Correspondence Withe Total Metabolic Rate

Based on the discussion in the previous section, let us begin by calculating the heat capacity using the results obtained in Section 2.

The heat capacity in solids is defined as:(44)$$C_{v}=\frac{dQ}{dT}$$
where *Q* is the amount of heat that must be added to a object with a mass of M in order to raise its temperature by *T*.

Then, by considering that
(45)$$J_{heat}=\frac{\dot{Q}}{A}=\frac{1}{A}\frac{dQ}{dt},$$
substituting Equation (Equation 45) into Equation (Equation 44) yields:(46)$$C_{v}=\frac{J_{heat}}{A\frac{dT}{dt}}$$
where $J_{heat}$ is the heat flux.

In our case, taking into account Equation (Equation 2) under isothermal conditions and considering the kinetic coefficients and the transport parameters defined by Goupil et al. [32], we can rewrite Equation (Equation 46) as:(47)$$C_{v}=\frac{e^{2}TS_{N}\sigma_{T}\nabla \mu_{e}}{A\frac{dT}{dt}}$$
where *e* is the particle’s charge, *T* denotes the absolute temperature, $S_{N}$ is the entropy transported per carrier (or per particle) as defined by Callen [48], $\sigma_{T}$ is the isothermal electrical conductivity, and $\Delta \mu_{e}$ is the electrochemical potential.

Now, as performed by Demtrius et al. [46], we consider that in the steady state, the energy (*u*) associated with ADP phosphorylation is given by $u=TS_{N}$, where $S_{N}$, in our case, denotes the thermodynamic entropy of the cell and (*T*) denotes the absolute temperature; then, Equation (Equation 47) becomes:(48)$$C_{v}=\frac{e^{2}u\sigma_{T}\nabla \mu_{e}}{A\frac{dT}{dt}}$$

Then, assuming that the density of the cell is uniform and that the entropy is proportional to the total volume of the cell, i.e., $u=aW_{c}$, where $W_{c}$ denotes the cell size and *a* a is the proportionality constant, the last expression into Equation (Equation 48) yields:(49)$$C_{v}=\frac{e^{2}aW_{c}\sigma_{T}\nabla \mu_{e}}{A\frac{dT}{dt}}$$

Based on Equation (Equation 49), let us assume that $A\frac{dT}{dt}\propto \tau_{c}$, i.e, the temperature (more precisely, the energy) under a steady state depends on the mean turnover time of the oxidation–reduction reaction, which is the metabolic cycle ($\tau_{c}$). Thus, Equation (Equation 49) becomes:(50)$$C_{v}=\frac{e^{2}aW_{c}\sigma_{T}\nabla \mu_{e}}{\gamma \tau_{c}}$$
where γ is a proportionality constant.

Demtrius et al. [46] derived an expression for the metabolic time ($\tau_{c}$) in the two regimes for long and short cycle times; they associated these times with different types of organisms, namely (a) systems with a large cell size, typically green plants whose cells contain chloroplasts as energy-transducing organelles and (b) systems with a relatively small cell size. Therefore, these times are:(51)$$\tau_{c}=\xi W_{c}^{\frac{\left(1-\mu \right)}{\mu }}$$
and
(52)$$\tau_{c}=\xi W_{c}^{\frac{\left(1\right)}{4\mu }}$$
where $W_{c}$ denotes the cell size, μ is the metabolic efficiency (closely related to the degree of coupling between the electron transport process and ADP phosphorylation), and ξ is a constant. Then, if we substitute these expressions into Equation (Equation 49), we obtain:(53)$$C_{v}=\frac{e^{2}a}{\xi }W^{2-\frac{1}{\mu }}\sigma_{T}\nabla \mu_{e}$$
and
(54)$$C_{v}=\frac{e^{2}a}{\xi }W^{4-\frac{1}{\mu }}\sigma_{T}\nabla \mu_{e}$$

According to Equations (53) and (54), the correspondence between the total metabolic rate (*P*) (see Equation (43)) and the heat capacity ($C_{v}$) is clear.

### 3.4. Different Thermodynamic Work Regimes in the Context of the Allometric Laws

Now, when applying Ohm’s law to the proton circuit, since for the isothermal process, an electric field derived from the electrochemical potential is generated in such a manner that is in accordance with Equations (3) and (29), the electrochemical potential becomes:(55)$$\nabla \mu_{e}=-\alpha_{g}IR_{in}$$

As mentioned in Section 2, we have three working regimes: (a) maximum power (MPO), (b) maximum ecological function (MEF), and (c) maximum power efficiency (MPE). Depending on which one we choose, a maximum current can be achieved. Thus, we obtain the heat capacities depending on the regime on which we are working as follows.

#### 3.4.1. Maximum Power Regime

For this working regime, substituting Equation (Equation 36) into Equation (Equation 55) yields:(56)$$\nabla \mu_{e}=\frac{\alpha_{g}}{2}V_{0}$$

Then, by entering Equation (Equation 56) into Equations (53) and (54), the heat capacity becomes:(57)$$C_{v}=\left(\frac{1}{2}\right)\frac{e^{2}a\alpha_{g}}{\xi }W^{2-\frac{1}{\mu }}\sigma_{T}V_{0}$$
and
(58)$$C_{v}=\left(\frac{1}{2}\right)\frac{e^{2}a\alpha_{g}}{\xi }W^{4-\frac{1}{\mu }}\sigma_{T}V_{0}$$

#### 3.4.2. Ecological Function Regime

For this working regime, substituting Equation (Equation 38) into Equation (Equation 55) yields:(59)$$\nabla \mu_{e}=\frac{3\alpha_{g}}{4}V_{0}$$

Then, entering Equation (Equation 56) into Equations (53) and (54), the heat capacity becomes:(60)$$C_{v}=\left(\frac{3}{4}\right)\frac{e^{2}a\alpha_{g}}{\xi }W^{2-\frac{1}{\mu }}\sigma_{T}V_{0}$$
and
(61)$$C_{v}=\left(\frac{3}{4}\right)\frac{e^{2}a\alpha_{g}}{\xi }W^{4-\frac{1}{\mu }}\sigma_{T}V_{0}$$

#### 3.4.3. Maximum Efficient Power Regime

For this working regime, substituting Equation (Equation 40) into Equation (Equation 55) yields:(62)$$\nabla \mu_{e}=\frac{2\alpha_{g}}{3}V_{0}$$

Then, entering Equation (Equation 56) into Equations (53) and (54), the heat capacity becomes:(63)$$C_{v}=\left(\frac{2}{3}\right)\frac{e^{2}a\alpha_{g}}{\xi }W^{2-\frac{1}{\mu }}\sigma_{T}V_{0}$$
and
(64)$$C_{v}=\left(\frac{2}{3}\right)\frac{e^{2}a\alpha_{g}}{\xi }W^{4-\frac{1}{\mu }}\sigma_{T}V_{0}$$

For the special case in which the scaling exponent is β=2/3 (mammals and large birds), Figure 3 depicts the heat capacity ($C_{v}$) as a function of the organism size (*W*) for the three different working regimes.

## 4. Discussion and Concluding Remarks

In this paper, the methodology of so-called LIT is applied to the well-known phenomenon of thermoelectricity. LIT is traditionally used locally to study general systems in non-equilibrium states that are considered both internal and external contributors to entropy increments in order to analyze the efficiency of two coupled processes with generalized fluxes ($J_{1}$ and $J_{2}$) and their corresponding forces ($X_{1}$ and $X_{2}$). We extend the former to analyze thermodynamic properties: the power output, efficiency, and entropy production, through which energy is exchanged in a thermoelectric device but from the point of view of non-equilibrium thermodynamics. By using linear irreversible thermodynamics, it is possible to analyze three different regimes of operation: maximum power output (MPO), maximum ecological function (MEF), and maximum efficient power function (MEPF).

By considering thermoelectric devices as systems that can work under different performance regimes in order to optimize either power output, entropy production, or a mixture of these quantities, it is possible to obtain the maximum power output and the maximum electric current of a thermoelectric device known as a thermoelectric generator (TEG) in the so-called maximum power regime (see, for instance, Equations (36) and (37)), which corresponds to the values obtained previously for a thermoelectric generator but using a different approach [32].

Afterward, we obtained the maximum power output and the maximum electric current but under different working regimes (see Equations (38) and (39) for the ecological function regime and Equations (40) and (41) for the maximum efficient power regime). According to Figure 1, it is clear that $P_{MPO}>MEP>MEF$ when ΔT→1, but $P_{MPO}\;\approx \;MEP\approx MEF$ when ΔT→0. As is well known for all heat engines such as thermoelectric generators, maximum power generation efficiency is thermodynamically limited by the Carnot efficiency ($\frac{\Delta T}{T_{h}}$); for the different devices that use the Seebeck effect for the generation of electricity, there are different output powers (see, for instance, Equations (37), (39), and (41)) depending on the ΔT that these devices can handle and the working regime. Therefore, the lesson is very clear: in order to achieve better maximum power in a thermoelectric device, it is necessary to have a ΔT close to 1, which necessarily impacts the efficiency of the device once the upper bound is the Carnot efficiency. Moreover, across a large ΔT, the electrical current required for the highest-efficiency operation changes as the material’s properties change with temperature or segment [49]; the latter imposes an additional requirement related to thermoelectric materials, which must depend on the thermoelectric parameters, as per the Seebeck coefficient, the conductivity of the materials, and its figure of merit (*Z*).

For instance, the highest *Z* that has been achieved for inorganic materials is 2.2, whereas for organic materials, the highest *Z* is 0.75 [50]. The value of *Z* is mainly due to low thermal conductivity and high electrical conductivity, which assist in the transfer of electrons during the thermoelectric process [51]. In some cases, it is not the figure of merit but the Seebeck coefficient (measured by dividing the difference in voltage at room temperature and the required temperature by the difference between these temperatures $S=\frac{-\Delta V}{\Delta T}$) that is being improved in order to obtain better efficiency [52]. It is clear from Figure 1 that in order to achieve better maximum power (which is related to *Z*) in a thermoelectric device, it is necessary to have a ΔT close to 1.

It is important to comment that new findings seem to show that the efficiency in the conversion of heat into electricity is proportional to $\Delta T^{2}$ instead of ΔT. The main reason for this is that in real technological devices, the heat source frequently fluctuates, so it is necessary to introduce a kind of transient heat source instead of the traditional steady-state heat flux, with the latter producing a change in the figure of merit that is larger in the transient state than in the steady state [53].

On the other hand, according the empirical studies conducted by Kleiber in 1961 [46], analyzing the exchange of energy in living beings, two elements are characterize the behavioral and physiological properties of an organism: its body size and its metabolic rate. Therefore, by considering the fact that the metabolism of an organism is generated by the energy released by electrons and protons as the particles are transferred from donor to acceptor states within the energy-transducing membrane [46], it is possible to establish a relation between the thermodynamic variables and the metabolic parameters, in particular, between the heat capacity and the metabolic rate. By carrying this out, it is possible to obtain the heat capacity for different work regimes using linear irreversible thermodynamics. Since there is a correspondence between the metabolic rate and heat capacity, different relationships for the heat capacities or metabolic rates can be established, depending on the work regime in question.

The algebraic structure of the obtained expressions (Equations (57), (58), (60), (61), (63) and (64)) are similar to the expressions obtained by Demetrius et al. [46] for allometric laws, depending on the phylogenetic status (plants or mammals), particularly if there is an agreement between the β exponent in the allometric laws (Equation (Equation 43)); however, there is no agreement between the constants of proportionality in the same equation.

In Table 1, particularly, in Figure 3, we can observe that in the expressions representing the metabolic rate, the constant factors change depending on the working regimen; the above might be related to the empirical observations made by Hemmingsen [30] on unicellular organisms, poikilotherms, and homeotherms, in which, when relating the metabolic rate to the size of the bodies of organisms, this produced a similar scaling exponent but with very different proportionality constants in the studied cases, i.e., the proportionality constant changes depending on the phylogenetic organism because of the working regime in which these "heat engines" are working. Moreover, the efficiency with which the energy is transduced depends, again, on the working regime used, generating different proportionality constants for distinct organisms, in concordance with the observation made by Tilman et al. (2004) [46,54], i.e., that many different ecological roles can be performed by organisms of a similar size and temperature.

## Figures and Tables

**Figure 1 entropy-25-01575-f001:**
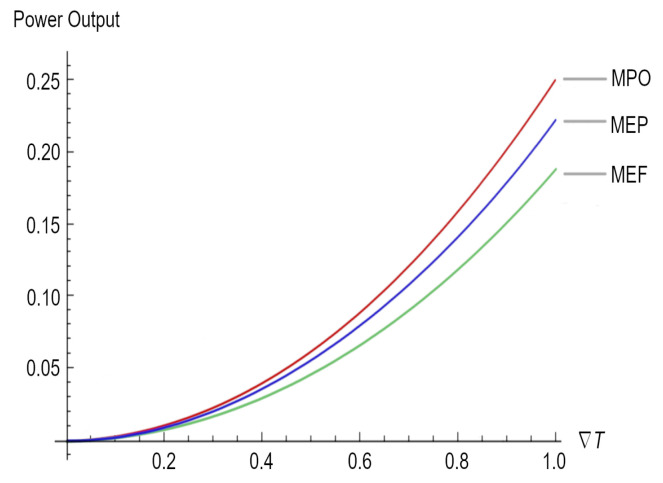
Power output for the three regimes, namely maximum power output (MPO), maximum ecological function (MEF), and maximum efficient power (MEP).

**Figure 2 entropy-25-01575-f002:**
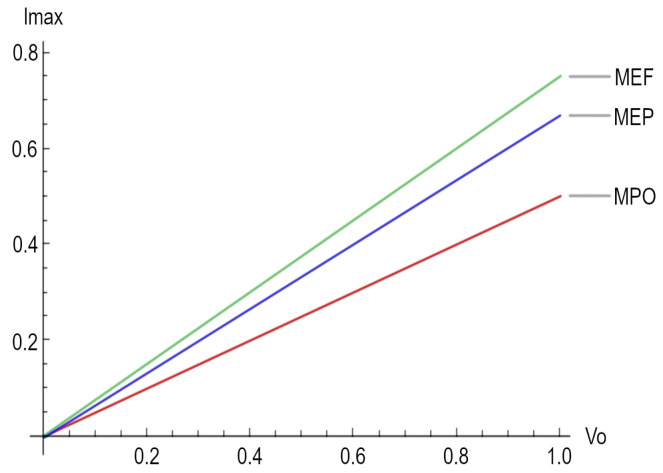
Maximum electrical current for the three regimes, namely maximum power output (MPO), maximum ecological function (MEF), and maximum efficient power (MEP).

**Figure 3 entropy-25-01575-f003:**
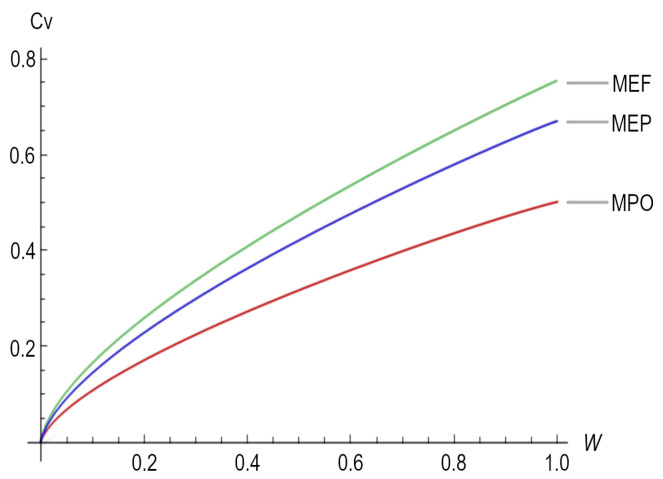
Heat Capacity ($C_{v}$) as a function of the organism size (*W*) for the three regimes, namely maximum power output (MPO), maximum ecological function (MEF), and maximum efficient power (MEP).

**Table 1 entropy-25-01575-t001:** Heat Capacity ($C_{v}$) as a function of the organism size (*W*) for the three different working regimes.

Working Regime	Plants	Mammals
MPO	$C_{v}$ = $\left(\frac{1}{2}\right)\frac{e^{2}a\alpha_{g}}{\xi }W^{2-\frac{1}{\mu }}\sigma_{T}V_{0}$	$C_{v}=\left(\frac{1}{2}\right)\frac{e^{2}a\alpha_{g}}{\xi }W^{4-\frac{1}{\mu }}\sigma_{T}V_{0}$
MEF	$C_{v}=\left(\frac{3}{4}\right)\frac{e^{2}a\alpha_{g}}{\xi }W^{2-\frac{1}{\mu }}\sigma_{T}V_{0}$	$C_{v}=\left(\frac{3}{4}\right)\frac{e^{2}a\alpha_{g}}{\xi }W^{4-\frac{1}{\mu }}\sigma_{T}V_{0}$
MEPF	$C_{v}=\left(\frac{2}{3}\right)\frac{e^{2}a\alpha_{g}}{\xi }W^{2-\frac{1}{\mu }}\sigma_{T}V_{0}$	$C_{v}=\left(\frac{2}{3}\right)\frac{e^{2}a\alpha_{g}}{\xi }W^{4-\frac{1}{\mu }}\sigma_{T}V_{0}$

## Data Availability

Not applicable.

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
