# Peer review of "Linear Irreversible Thermodynamics: A Glance at Thermoelectricity and the Biological Scaling Laws"

_entropy, 2023, doi:10.3390/e25121575_

Round 1

Reviewer 1 Report

Comments and Suggestions for Authors

see attached file

Reviewer 2 Report

Comments and Suggestions for Authors

This work provides a new insight into thermoelectrics using the theory of linear irreversible thermodynamics, which may raise significant interest among readers. It can be accepted before the following concerns are addressed.

1. Did you only consider the steady state? Could the linear irreversible thermodynamics be extended from a steady state to a transient one? At least the corresponding statements about the dynamics of the thermoelectric should be provided in the context and the literature "Energy Convers Manage 2023;296:117669" and "Int J Heat Mass Transfer 2023;211:124203" may provide references for authors.

2. The quality of the figures should be improved.

3. In section 3, it is suggested to add the corresponding data (in the form of figure) to support your findings.

4. As is well known, the finite element method (FEM) is widely used to analyze the performance of thermoelectric devices, such as "Energy. 2022;238:121816". What is the difference between FEM and LIT? and which one is more reasonable?

Comments on the Quality of English Language

Minor editing of English language required

Reviewer 3 Report

Comments and Suggestions for Authors

The authors make one of the rare attempts to explain biological laws such as the allometric law in physical terms. Such attempts should be made much more, which is why I was willing to support the manuscript with my expert opinion.
Unfortunately, in my understanding, the article did not achieve an empirical explanation for allometric laws. Which is why I support the rejection of the manuscript and would like to ask the authors to present their thoughts more clearly. I also miss the comparison of thermodynamic predictions and experimental findings (e.g. the scaling exponent).

The article begins with textbook knowledge of the linear irreversible thermodynamics of electrothermal effects.

Functions such as the Maximum Ecological Function are introduced and not explained. 

A list of all symbols would be helpful. What is meant by particle charge? The elementary charge, the charge of a proton??? To what is the electrochemical potential related?

The ref. in line 195 is wrong.

Round 2

Reviewer 1 Report

Comments and Suggestions for Authors

The authors revised the paper.

Reviewer 2 Report

Comments and Suggestions for Authors

This work can be accepted now.

Reviewer 3 Report

Comments and Suggestions for Authors

Thanks for answering my questions.